# Nociceptin/Orphanin FQ Opioid Peptide-Receptor Expression in the Endometriosis-Associated Nerve Fibers—Possible Treatment Option?

**DOI:** 10.3390/cells12101395

**Published:** 2023-05-15

**Authors:** Qihui Guan, Renata Voltolini Velho, Alice Jordan, Sabrina Pommer, Irene Radde, Jalid Sehouli, Sylvia Mechsner

**Affiliations:** Endometriosis Research Center, Department of Gynecology Charité with Center of Oncological Surgery, Charité, Campus Virchow-Klinikum, Augustenburger Platz 1, 13353 Berlin, Germany; qi-hui.guan@charite.de (Q.G.); renata.voltolini-velho@charite.de (R.V.V.);

**Keywords:** chronic inflammation, endometriosis, nerve fibers, NOP, opioid receptors, pelvic pain

## Abstract

Endometriosis (EM) is a chronic inflammatory disease affecting millions of women worldwide. Chronic pelvic pain is one of the main problems of this condition, leading to quality-of-life impairment. Currently, available treatment options are not able to treat these women accurately. A better understanding of the pain mechanisms would be beneficial to integrate additional therapeutic management strategies, especially specific analgesic options. To understand pain in more detail, nociceptin/orphanin FQ peptide (NOP) receptor expression was analyzed in EM-associated nerve fibers (NFs) for the first time. Laparoscopically excised peritoneal samples from 94 symptomatic women (73 with EM and 21 controls) were immunohistochemically stained for NOP, protein gene product 9.5 (PGP9.5), substance P (SP), calcitonin gene-related peptide (CGRP), tyrosine hydroxylase (TH), and vasoactive intestinal peptide (VIP). Peritoneal NFs of EM patients and healthy controls were positive for NOP and often colocalized with SP-, CGRP-, TH-, and VIP-positive nerve fibers, suggesting that NOP is expressed in sensory and autonomic nerve fibers. In addition, NOP expression was increased in EM associate NF. Our findings highlight the potential of NOP agonists, particularly in chronic EM-associated pain syndromes and deserve further study, as the efficacy of NOP-selective agonists in clinical trials.

## 1. Introduction

Endometriosis (EM) is an unrecognized, chronic inflammatory gynecological disease that affects approximately 10% of women of reproductive ages, i.e., two million women in Germany and 270 million worldwide [1,2]. Characterized by the presence of epithelial, stromal, and muscle cells outside the uterine cavity, EM affects the uterus itself (adenomyosis uteri) and the peritoneum of the pelvic cavity. Several factors interplay in the genesis of EM-associated pain such as the lesions themselves, pain-mediating substances, nerve fibers, and immune cells [2]. Due to the dissemination of endometriotic lesions, the associated symptoms show a wide variation including chronic pelvic pain, dysmenorrhea, dyspareunia, dyschezia, dysuria, and sub- or infertility [1,2,3,4]. Chronic pelvic pain is one of the main problems of this condition, affecting patients’ psychological and social wellbeing and imposing a substantial economic burden on society [5,6,7,8,9].

Currently, the standard treatment options are not able to solve the daily problems of these women accurately [2]. Hormonal administration and surgical intervention are the most applied treatment options; however, countless side-effects, as well as high recurrence rates and ongoing pain after intervention, are frequent [2,3]. As pain is an important factor in EM, analgesia should be applied. To date, nonsteroidal anti-inflammatory drugs (NSAD), metamizole, or, in extreme cases, opioids are the drugs used for EM-associated pain treatment [2]. Clinical observations often show NSAD and metamizole failure in the case of ongoing symptoms over many years and chronic pelvic pain syndrome, implying that central sensitization seems to be part of the pain chronification leading to a decrease in the pain threshold [2,4,10]. Therefore, the development of more effective therapeutic strategies is still an unmet clinical need, and it is hindered by the lack of knowledge of the mechanisms underlying the generation of EM pain and its associated comorbidities.

Opioid receptors are membrane-bound receptors belonging to the family of G-protein-coupled receptors (GPCRs). There are four opioid receptor subtypes, including the three classical opioid receptors, µ (MOR), δ (DOR), and κ (KOR), and the more recently discovered nociceptin/orphanin FQ peptide (NOP) receptor [11,12]. Since NOP receptors are distributed in various regions (dorsal root ganglia—DRG, spinal dorsal horn—SDH, and brain) that are involved in pain transmission, they are under investigation primarily as alternatives for MOP receptor opioid analgesics, in addition to their anxiolytic and antidepressant-like effect [13]. However, in the earlier phases of the discovery of nociception, the NOP receptor was considered a controversial drug target for analgesics because of its unique pharmacological effects on pain modulation (antinociceptive vs. nociceptive effects) [14,15,16,17,18,19,20]. Currently, the NOP receptor has become the main focus as a promising target for analgesics as NOP receptor ligands have been reported to show antinociceptive effects in nonhuman primates regardless of their administered doses and administration routes. Moreover, NOP/opioid receptor agonists have recently displayed potent antinociceptive activity with favorable side-effect profiles [13].

To understand pain generation in EM patients in more detail, NOP expression was analyzed for the first time in EM-associated nerve fibers (NF).

## 2. Materials and Methods

### 2.1. Patients

This prospective study enrolled 94 women from May 2012 until May 2019. Seventy-three EM patients, who underwent laparoscopy due to symptomatic EM with excision of endometriotic lesions, were included. The peritoneal lesions were localized in the lateral pelvic wall (*n* = 17), bladder (*n* = 4), pouch of Douglas (*n* = 14), uterosacral ligament (*n* = 6), peritoneum (*n* = 2), and fossa ovarica (*n* = 30). The diagnosed EM was staged according to the revised classification of the American Society of Reproductive Medicine (rASRM) as (I) minimal, (II) mild, (III) moderate, and (IV0 severe. In the analysis, two stages were considered: mild (rASRM I and II) and severe (rASRM III and IV). Twenty-one control samples were collected from women without EM, who had undergone laparoscopy for benign gynecological presentations such as non-EM associated with ovarian cysts, uterine fibroids, *Hydrosalpingx*, pelvic pain, peritonealized tissue, or the unfulfilled wish to have children. Additionally, clear peritoneal fluids were obtained during laparoscopy from patients with peritoneal EM (*n* = 17) and controls (*n* = 17).

Patients were selected on the basis of clinical intraoperative and subsequent histopathologic findings. All patients were given a complete gynecological examination including palpation and transvaginal ultrasound. The severity of pain was documented using a standardized questionnaire with a visual analogue scale (VAS; 0 = no pain, 10 = strongest imaginable pain) [21,22,23]. The women were divided into two groups according to the pain scale: moderate pain (0–5 on the scale) and severe pain (6–10 on the scale).

The study was approved by the Institutional Review Board of the Charité University Medical Centre (Ethic vote EA4/036/12). All patients gave their consent.

### 2.2. Immunofluorescence Double Staining and Determination of Nerve Fiber Density

All peritoneal biopsies were immediately fixed in buffered formalin (4%) for at least 12 h and thereafter embedded in paraffin. Sections of 2 µm thickness were cut and used for immunofluorescence double staining using antibodies against NOP receptor (Santa Cruz, Heidelberg, Germany, sc-398073, 1:50 and Abcam, Cambridge, UK, ab66219, 1:380), protein gene product 9.5 (PGP 9.5—Novus, Wiesbaden Nordenstadt, Germany, NB110-58872, 1:300), substance P (SP, Santa Cruz, sc-21715, 1:500), calcitonin gene-related peptide (CGRP—Santa Cruz, sc-8857, 1:100), tyrosine hydroxylase (TH—Sigma, St. Louis, MO, USA, T2928, 1:100), and vasoactive intestinal peptide (VIP—Santa Cruz, sc-25347, 1:100).

Negative control sections were processed by omitting the specific primary antibody. A skin incision and a tissue section of peritoneal EM with large nerve incisions were used as the positive control. Staining was detected using an axiophot (Carl Zeiss, Göttingen, Germany) microscope. Photomicrographs were taken at different magnifications (100× and 200×) and were further processed using Adobe Photoshop (2022 Full Version, cs6, Adobe Systems, Unterschleissheim, Germany).

The density of PGP9.5 nerve fibers was assessed by counting the number of immunostained nerves proximal to the endometriotic lesions (epithelial, stromal, and smooth muscle cells) and in the distal area at 1 mm^2^. The “hotspot” method [24] was used to determine the nerve fiber density of the control tissue as already described.

The density was measured by sequential assessment of two blinded investigators. Each patient had a code, which was unbroken until after the analysis at the end of the study. In cases of discrepant results, both the first and the second observers repeated the analysis together and reached a consensus.

### 2.3. Enzyme-Linked Immunosorbent Assay

Peritoneal fluids were aspirated from the pouch of Douglas immediately after the insertion of trocars to minimize contamination with blood. Grossly hemorrhagic specimens were excluded. Peritoneal fluids were centrifuged for 5 min at 3000 rpm, and the supernatants were aliquoted and stored at −80 °C until used. The endogenous Orphanin FQ/Nociceptin ligand concentration was measured in duplicate using the commercially available Human Orphanin FQ/Nociceptin enzyme-linked immunosorbent assay (ELISA) kit (EKH6946—Nordic BioSite AB, Täby, Sweden). This kit presents a detection range of 4.688–300 pg/mL and a sensitivity of 2.813 pg/mL. The analysis was conducted according to the manufacturer’s protocol. After the substrate reaction, the optical density was measured (absorbance at 450 nm) automatically by the ELISA-READER Thermo Scientific Multiskan FC (Waltham, MA, USA, Unity Lab Services).

### 2.4. Statistical Analysis

Statistical analysis was performed using IBM SPSS for Windows (version 29.0.0.0). The data were evaluated using *t*-test (parametric) or Mann–Whitney U, Kruskal–Wallis, and Spearman correlation tests (nonparametric). Chi-square and Fisher’s exact tests were used for the qualitative variable. Statistical significance was defined for *p* < 0.05.

## 3. Results

### 3.1. Population Characteristics

The population characteristics including pain aspects from the 94 women recruited for this study are summarized in Table 1. The present study group comprised 73 patients: 50 (68.49%) presented with minimal to mild endometriosis (rASRM I and II) and 23 (31.51%) presented with moderate to severe endometriosis (rASRM III and IV). Twenty-two (22/73) of them were under hormonal therapy at the time of the surgery. The mean age of the EM patients was 31.2 (18–50) years. The control group was a composite of 21 patients, two of which received hormonal therapy. Women in the control group were on average 35.6 (18–52) years old. No significant difference in age between EM and non-EM patients was observed (*p* = 0.131).

### 3.2. Characterization of Nerve Fibers in Peritoneal Endometriotic Lesions

Using anti-PGP9.5, nerve fibers could be detected in both EM and healthy peritoneal specimens. PGP9.5 nerve fiber density was significantly increased in endometriotic lesions (mean ± SD: 1.76 ± 1.57 NF/mm^2^) compared to the healthy peritoneum (mean ± SD: 0.28 ± 0.70 NF/mm^2^; *p* < 0.001). When the hormonal therapy was taken into consideration, a decreased innervation could be observed in the EM group with hormonal intake (mean ± SD: 1.05 ± 0.79 NF/mm^2^) compared with the EM group without hormonal treatment (mean ± SD: 2.15 ± 1.76 NF/mm^2^; *p* = 0.011) (Figure 1A–C).

A correlation between the nerve density (PGP9.5—positive NF/mm^2^) and the rASRM stages could be seen (r = 0.403; *p* < 0.001) (Table 2).

### 3.3. EM Patients Showed Increased Expression of NOPReceptor

EM patients presented more NOP-positive nerve fibers (mean ± SD: 1.22 ± 1.62 NOP-positive NF/mm^2^) when compared with controls (mean ± SD: 0.11 ± 0.17 NOP-positive NF/mm^2^; *p* < 0.001). Interestingly, EM patients that received hormonal therapy did not differ in the expression of NOP receptor (mean ± SD: 0.83 ± 0.99 NOP-positive NF/mm^2^) from the patients that did not receive this treatment (mean ± SD: 1.49 ± 1.86 NOP-positive NF/mm^2^; *p* = 0.381) (Figure 1D,E).

When looking at blood vessels, EM patients presented more blood vessels (mean ± SD: 6.0 ± 4.7 blood vessels/mm^2^) than women without EM (mean ± SD: 2.7 ± 4.3 blood vessels/mm^2^; *p* = 0.007). In addition, EM patients showed more NOP-positive stained vessels (mean ± SD: 1.1 ± 1.8 blood vessels NOP-positive/mm^2^) than the control group (mean ± SD: 0.2 ± 0.7 blood vessels NOP-positive/mm^2^; *p* = 0.013) (Figure 1F,G). The hormonal intake did not affect the number of blood vessels or their NOP-positivity.

The NOP-positive stained nerve fibers (r = 0.410; *p* < 0.001) and blood vessels (r = 0.307; *p* = 0.024) correlated with the rASRM stages but not with the pain levels (Table 2).

### 3.4. NOP Receptors Are Located on Sympathetic, Parasympathetic, and Sensory Fibers That Innervate the Lesions

If the NOP receptor is involved in EM and its associated pain, this receptor should be located on the axonal fibers innervating the lesions. With double-labeling fluorescence immunohistochemistry, most (>75%) sympathetic fibers (TH-positive), many (50–75%) parasympathetic fibers (VIP-positive), and many (50–75%) sensory fibers (SP- and CGRP-positive) in the EM and control samples were co-labeled with an antibody for NOP receptor (Figure 2).

### 3.5. Orphanin FQ/Nociceptin Ligand Is Not Overexpressed in the Peritoneal Fluid of Women with Peritoneal Endometriosis

The endogenous Orphanin FQ/Nociceptin ligand concentration expression in the peritoneal fluid of women with peritoneal EM (mean ± SD: 2.81 ± 11.59) and controls (7.88 ± 22.07) was not statistically different (*p* = 0.586; Figure 3). All EM women were premenopausal with a mean age of 31.4 ± 4.4 years (range 26–40 years) and had a regular menstrual cycle (secretory phase: 41%, proliferative phase: 59%). EM was classified according to the rASRM at stages I to IV (I = 29.41%, II = 35.29%, III = 00.00%, IV = 35.29%). Controls included women who were premenopausal with a mean age of 32.6 ±.5 years (range 20–50 years) and had a regular menstrual cycle (secretory phase: 29.4%, proliferative phase: 41.2%, menses: 5.9%), while one woman was taking an oral contraceptive. 

## 4. Discussion

There are many symptoms connected with EM. However, the main symptom is cyclic and noncyclic chronic pelvic pain [2,25]. The pain pathology is still largely unexplained; however, since the discovery of the NOP receptor and N/OFQ as the endogenous ligand, evidence has appeared demonstrating the involvement of this receptor system in pain. This is not surprising for members of the opioid receptor and peptide families, particularly since both the receptor and N/OFQ are highly expressed in brain regions involved in pain, as well as in the spinal cord and dorsal root ganglia [26,27]. Moreover, most of the data on NOP receptor expression are derived from rodents, and using mRNA expression analysis which may not translate in the human scenario [28]. NOP is expressed both in the central nervous system and in peripheral tissues. Nevertheless, little is known about the localization of the NOP receptor in human tissues, and information about any changes in expression levels in human disease is absent [29]. Whole-body images of a healthy 22 year old man showed radioactivity of ^11^C-NOP-1A in the brain and peripheral organs expressing NOP receptors, such as the heart, lungs, liver, pancreas, small bowel, and urinary bladder [30]. In human visceral disease, we could only find data about NOP expression in bladder pain syndrome [30]. Accordingly, we analyzed, for the first time, the nociceptin/orphanin FQ peptide receptor expression in EM-aNF. Our goal was to understand pain generation in EM patients in more detail, relating this pain to the localization and expression of NOP receptors in nerve fibers from the female reproductive system and visceral organs.

The hyper-innervation already described in endometriotic lesions [31,32,33,34,35,36] was confirmed in the 73 EM women enrolled in this study. Interestingly, the NOP receptor was significantly more expressed in nerve fibers and blood vessels from EM patients than in controls. This is substantial evidence for peripheral sensitization and involvement of EM-aNF in pain generation. A marked and significant increase in NOP receptor immunoreactive nerve fibers was observed in bladder specimens from patients with overactive bladder and with bladder pain syndrome—another chronic pelvic pain condition [29]. Subclassification of the EM-aNF quality showed colocalization of the NOP receptor in sensory (SP- and CGRP-positive NF) and autonomic NF (TH- and VIP-positive NF), which have been seen for other groups [37], demonstrating the complexity of the EM-associated pain.

The Involvement of the NOP receptor system in pain modulation has been carefully investigated [37,38]. Indeed, depending on route, concentration, and pain model, NOP receptor activation could lead to either pronociceptive or antinociceptive effects [38,39]. The effects of NOP receptor agonist activation appear to be considerably clearer for chronic than acute pain [37,38]. Early studies examining the effects of N/OFQ on pain induced by inflammation or sciatic nerve injury suggested potential neuroplasticity, as the peptide was very effective in inducing anti-allodynic and anti-hyperalgesic activity in these chronic pain models [40,41,42,43].

EM-associated pain has a strong impact on the patient’s quality of life [44,45]. Unfortunately, current treatment strategies are not fully satisfactory [2]; thus, novel treatments with better effectiveness and tolerability are urgently needed. Our study demonstrated a link among NOP receptor expression, rASRM, and pain in EM patients, suggesting that the NOP receptor and N/OFQ as the endogenous ligand may be involved in EM-associated pain. Further investigation will be needed to elucidate these links and to evaluate whether the NOP receptor could provide a target model for new therapeutic intervention.

## Figures and Tables

**Figure 1 cells-12-01395-f001:**
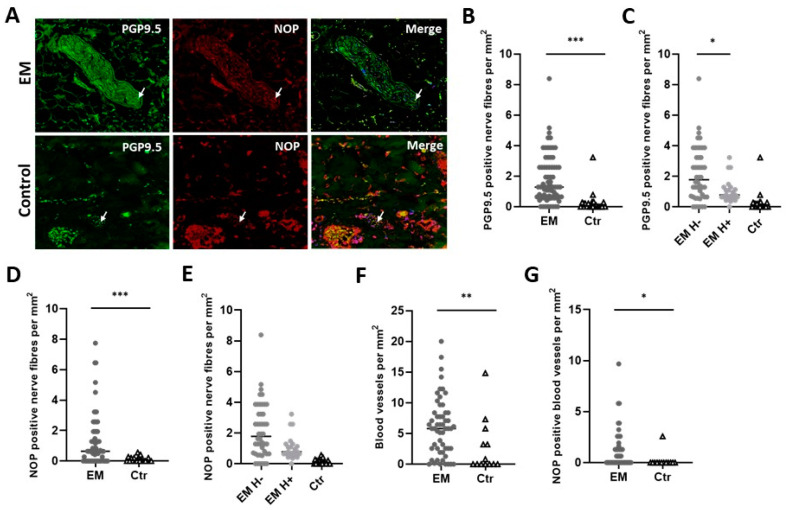
Endometriosis patients showed increased nerve fiber and blood vessel density, as well as NOP−receptor expression. (**A**) EM and control samples stained with PGP9.5 (green) and NOP (red) antibody. Merged images showi the colocalization of pan marker (PGP9.5) and the NOP receptor. All pictures are at 200× magnification. (**B**,**C**) PGP9.5−positive nerve fibers per mm^2^ in EM patients and controls. (**D**,**E**) NOP−positive nerve fibers per mm^2^ in EM patients and controls. (**F**) Blood vessels per mm^2^ in EM patients and controls. (**G**) NOP−positive blood vessels mm^2^ in EM patients and controls. Arrows indicate colocalization points. EM: endometriosis patients; Ctr: control; H+: under hormonal treatment; H−: without hormonal treatment. All results are presented as the median, and 25th–75th percentile. Mann–Whitney and Kruskal–Wallis with Dunn’s multiple comparison tests. * *p* < 0.05, ** *p* < 0.01, *** *p* < 0.001.

**Figure 2 cells-12-01395-f002:**
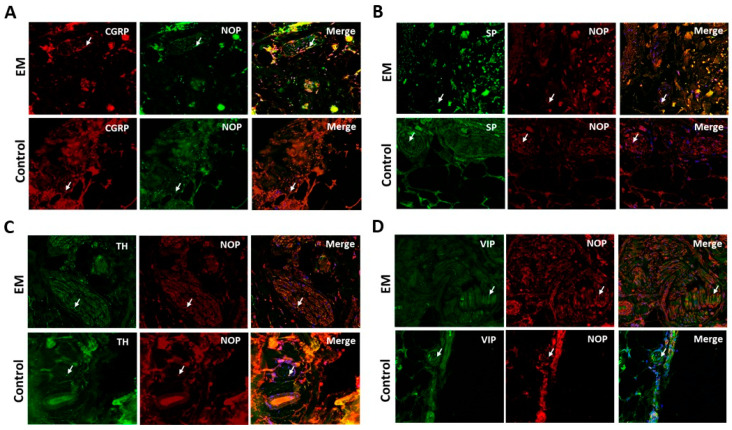
NOP receptors are present in sympathetic, parasympathetic, and sensory fibers innervating endometriotic lesions. (**A**) EM and control samples stained with CGRP (sensory fibers—red) and NOP (green) antibody. (**B**) EM and control samples stained with SP (sensory fibers—green) and NOP (red) antibody. (**C**) EM and control samples stained with TH (sympathetic fibers—green) and NOP (red) antibody. (**D**) EM and control samples stained with VIP (parasympathetic fibers—green) and NOP (red) antibody. Arrows indicate colocalization points between nerve marker and the NOP receptor. All pictures are 200× magnification.

**Figure 3 cells-12-01395-f003:**
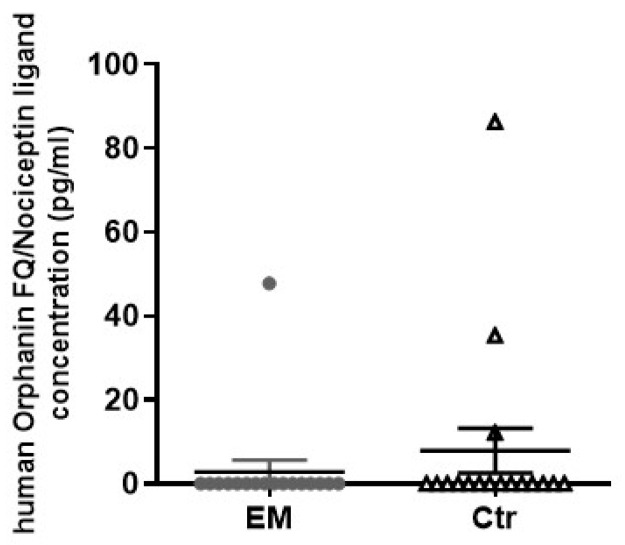
Orphanin FQ/Nociceptin ligand concentration (pg/mL) in the peritoneal fluid of women with peritoneal endometriosis and controls. Using an ELISA kit, the endogenous Orphanin FQ/Nociceptin ligand concentration expression in the peritoneal fluid of women with peritoneal EM and controls was measured. EM: endometriosis; Ctr: control. Mann–Whitney test; *p* = 0.586.

**Table 1 cells-12-01395-t001:** Characteristics of the study participants.

	EM Patients (N = 73)	Controls (N = 21)
Age (years)		
Mean	31.2	35.6
SD	6.93	10.65
Stages (rASRM)		
I–II	49 (67.1%)	-
III–IV	24 (32.9%)	-
Hormone treatment		
Yes	22 (30.1%)	2 (9.5%)
Missing data	5 (6.8%)	8 (38.1%)
Pain (EM-associated pain)		
Number of patients	70 (95.9%)	9 (42.8%)
Missing data	3 (4.1%)	11 (52.4%)
Pelvic pain		
Number of patients	66 (90.4%)	8 (38.1%)
Pain intensity (mean, SD)	5.27 ± 1.62	N.A.
Missing data	45 (61.6%)	8 (38.1%)
Dysmenorrhea		
Number of patients	64 (87.7%)	6 (28.6%)
Pain intensity (mean, SD)	5.59 ± 2.33	N.A.
Missing data	41 (56.2%)	6 (28.6%)
Dyspareunia		
Number of patients	47 (64.4%)	4 (19.0%)
Pain intensity (mean, SD)	4.64 ± 2.29	N.A.
Missing data	16 (%)	4 (19.0%)
Dyschezia		
Number of patients	25 (34.2%)	2 (9.5%)
Pain intensity (mean, SD)	4.55 ± 2.70	N.A.
Missing data	15 (20.5%)	2 (9.5%)
Dysuria		
Number of patients	11 (15.1%)	2 (9.5%)
Pain intensity (mean, SD)	3.25 ± 1.50	N.A.
Missing data	6 (8.2%)	2 (9.5%)
Menstrual cycle		
Menses	5 (6.8%)	0 (0.0%)
Proliferative	14 (19.2%)	1 (4.8%)
Secretory	11 (15.1%)	4 (19.0%)
Hormone intake	23 (31.5%)	2 (9.5%)
Menopause	0 (0.0%)	0 (0.0%)
Missing data	20 (27.4%)	14 (66.7%)

N.A.: no answer.

**Table 2 cells-12-01395-t002:** Correlation analysis.

			Value
Hormonal therapy in EM	Pelvic pain		*p* = 0.263 ^b^
	Dysmenorrhea		*p* = 0.599 ^b^
	Dyspareunia		*p* = 1.000 ^b^
	Dyschezia		x^2^ (1) = 0.512; *p* = 0.579 ^a^
	Dysuria		x^2^ (1) = 0.046; *p* = 1.000 ^a^
Pain level and hormonal therapy in EM	Pelvic pain		*p* = 0.674 ^b^
	Dysmenorrhea		x^2^ (1) = 0.022; *p* = 1.000 ^a^
	Dyspareunia		*p* = 0.388 ^b^
	Dyschezia		*p* = 0.543 ^b^
	Dysuria		*p* = 1.000 ^b^
Pain level and rASRM	Pelvic pain		*p* = 0.611 ^b^
	Dysmenorrhea		*p* = 1.000 ^b^
	Dyspareunia		*p* = 1.000 ^b^
	Dyschezia		*p* = 0.560 ^b^
	Dysuria		*p* = 0.405 ^b^
Pelvic pain and nerve fiber density/pain receptor	PGP9.5	Nerve fibers/mm^2^	r = 0.344; *p* = 0.108 ^c^
	NOP	Nerve fibers/mm^2^	r = −0.248; *p* = 0.253 ^c^
		Blood vessels/mm^2^	r = 0.067; *p* = 0.806 ^c^
Dysmenorrhea and nerve fiber density/pain receptor	PGP9.5	Nerve fibers/mm^2^	r = 0.142; *p* = 0.480 ^c^
	NOP	Nerve fibers/mm^2^	r = −0.142; *p* = 0.481 ^c^
		Blood vessels/mm^2^	r = −0.275; *p* = 0.270 ^c^
Dyspareunia and nerve fiber density/pain receptor	PGP9.5	Nerve fibers/mm^2^	r = 0.119; *p* = 0.475 ^c^
	NOP	Nerve fibers/mm^2^	r = 0.009; *p* = 0.958 ^c^
		Blood vessels/mm^2^	r = 0.69; *p* = 0.729 ^c^
Dyschezia and nerve fiber density/pain receptor	PGP9.5	Nerve fibers/mm^2^	r = −0.050; *p* = 0.740 ^c^
	NOP	Nerve fibers/mm^2^	r = 0.049; *p* = 0.742 ^c^
		Blood vessels/mm^2^	r = −0.032; *p* = 0.863 ^c^
Dysuria and nerve fiber density/nerve fiber receptor	PGP9.5	Nerve fibers/mm^2^	r = −0.060; *p* = 0.674 ^c^
	NOP	Nerve fibers/mm^2^	r = −0.152; *p* = 0.288 ^c^
		Blood vessels/mm^2^	r = −0.044; *p* = 0.801 ^c^
rARSM and nerve fiber density/nerve fiber receptor	PGP9.5	Nerve fibers/mm^2^	r = 0.403; *p* <0.001 **^,c^
	NOP	Nerve fibers/mm^2^	r = 0.410; *p* <0.001 **^,c^
		Blood vessels/mm^2^	r = 0.307; *p* = 0.024 *^,c^

Analyses using ^a^ chi-square or ^b^ Fisher and ^c^ Spearman correlation; * *p* < 0.05, ** *p* < 0.001.

## Data Availability

All data used and analyzed during the current study are available from the corresponding author upon reasonable request.

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
