# Peer review of "Nociceptin/Orphanin FQ Opioid Peptide-Receptor Expression in the Endometriosis-Associated Nerve Fibers—Possible Treatment Option?"

_cells, 2023, doi:10.3390/cells12101395_

Round 1

Reviewer 1 Report

 Qihui Guan and authors analyzed nerve fibers by determining the expression of NOP receptor, samples were collected from 73 females with established endometriosis and 21 controls. They  immunohistochemically stained for NOP, protein gene product 9.5 (PGP9.5), substance P (SP), calcitonin gene-related peptide (CGRP), tyrosine hydroxylase (TH) and vasoactive intestinal peptide (VIP). Authors verified higher expression of NOP in the patients with endometriosis and determined NOP often co-localized with the above mentioned nerve fiber markers.  

Minor comments

1) For better understanding of the readers please label or add into figure legends what the arrows indicate

2) Please elaborate into discussion what was the purpose to stain protein gene product 9.5 (PGP9.5), substance P (SP), calcitonin gene-related peptide (CGRP), tyrosine hydroxylase (TH) and vasoactive intestinal peptide (VIP) as there is no context to it apart from they were stained and colocalized. 

3) Supplementary data sheet can be easily merged into methods sections. 

Major comments

1) Authors do not provide any mechanism insights to target NOP using any animal model of endometriosis to develop additional therapeutic management strategies. 

2) It is not clear how the pain sensitivity was calculated between EM and controls as the pain intensity measurements are missing from most of the controls (Table 1). 

Author Response

Reviewer #1:

Qihui Guan and authors analyzed nerve fibers by determining the expression of NOP receptor, samples were collected from 73 females with established endometriosis and 21 controls. They  immunohistochemically stained for NOP, protein gene product 9.5 (PGP9.5), substance P (SP), calcitonin gene-related peptide (CGRP), tyrosine hydroxylase (TH) and vasoactive intestinal peptide (VIP). Authors verified higher expression of NOP in the patients with endometriosis and determined NOP often co-localized with the above mentioned nerve fiber markers. 

Minor comments

  1. For better understanding of the readers please label or add into figure legends what the arrows indicate

Answer: Thanks for your comment. Arrows indicate colocalisation points. We added this info in the legends.

  1. Please elaborate into discussion what was the purpose to stain protein gene product 9.5 (PGP9.5), substance P (SP), calcitonin gene-related peptide (CGRP), tyrosine hydroxylase (TH) and vasoactive intestinal peptide (VIP) as there is no context to it apart from they were stained and colocalized.

Answer: Thanks for your comment. We analysed for the first time, the nociceptin/orphanin FQ peptide receptor expression in EM-aNF. Our goal was to understand pain generation in EM patients in more detail, relating this pain with the localization and expression of NOP receptors in nerve fibers from the female reproductive system and visceral organs and the stainings were essential for that. We reorganize the discussion and added some more information that supports our study.

  1. Supplementary data sheet can be easily merged into methods sections.

Answer: Thanks for your comment. The supplementary data was merged into methods in lines 101-105.

Major comments

  1. Authors do not provide any mechanism insights to target NOP using any animal model of endometriosis to develop additional therapeutic management strategies.

Answer: Thanks for your comment. This was preliminary research on the NOP expression, localization, and correlation with EM-associated pain. To elucidate these links and understand the mechanism insights to target NOP, further investigations need to be carried on. Also, as EM only appears spontaneously in menstruating primates, the choice for animal model studies and the choice of the animal model itself, it´s something to be considered. Mimicking EM in mice brings advantages associated with low cost, ease to use, deep understanding of rodent biology, the possibility of using transgenic animals, and the ability to perform tests with a higher number of individuals. However, some drawbacks such as physiological differences (absence of a menstrual cycle, among others) and the phylogenetic distance between rodents and humans can be found. As a consequence, species-specific effects might appear in such a way that the effectiveness of some drugs tested in rodents (or even primates) may not translate to humans (Tejada et al., 2023 PMID: 36768741) as already observed with NOP receptor (Daibani et al., 2022 PMID: 35163856).

  1. It is not clear how the pain sensitivity was calculated between EM and controls as the pain intensity measurements are missing from most of the controls (Table 1).

Answer: Thanks for your comment. EM patients answered the VAS questionnaire, where the presence or absence of pain and pain intensity, could be accessed. In the control group, the presence or absence of pain was accessed through "yes or no" questions in a conversation with the physician. Questions as: “Do you have pelvic pain? Yes or no" were made for this group. No comparison between pain sensitivity or intensity was made between EM patients and controls. Most of the controls have no pain.

Reviewer 2 Report

Broad comments:

The article (cells-2353421) entitled “Nociceptin/orphanin FQ opioid peptide-receptor expression in the endometriosis-associated nerve fibers - possible treatment option? aims to understand pain in more detail, nociceptin/orphanin FQ peptide (NOP) receptor expression was analyzed in EM-associated nerve fibers (NF). The results highlight the potential of NOP agonists, particularly in chronic EM-associated pain syndromes and deserve further study, as the efficacy of NOP selective agonists in clinical trials. The research is generally valuable and well written.

The work is well written.

Author Response

The article (cells-2353421) entitled “Nociceptin/orphanin FQ opioid peptide-receptor expression in the endometriosis-associated nerve fibers - possible treatment option?” aims to understand pain in more detail, nociceptin/orphanin FQ peptide (NOP) receptor expression was analyzed in EM-associated nerve fibers (NF). The results highlight the potential of NOP agonists, particularly in chronic EM-associated pain syndromes and deserve further study, as the efficacy of NOP selective agonists in clinical trials. The research is generally valuable and well written.

Answer: Thank you for your comment.

Reviewer 3 Report

Thank you for the opportunity to evaluate this interesting manuscript. Every new potential biomarker in endometriosis diagnostics deserves special attention. 

This manuscript should be accepted after some minor revisions: 

- please include following references in Discussion section, with the aim to wide the overall knowledge about endometriosis pathophysiology (PMID: 36142815)

- please include recently published articles about endometriosis quality of life issues (PMID: 35819491) 

Author Response

Thank you for the opportunity to evaluate this interesting manuscript. Every new potential biomarker in endometriosis diagnostics deserves special attention.

This manuscript should be accepted after some minor revisions:

  1. please include following references in Discussion section, with the aim to wide the overall knowledge about endometriosis pathophysiology (PMID: 36142815)

Answer: Thank you for your comment. This reference was added.

  1. please include recently published articles about endometriosis quality of life issues (PMID: 35819491)

Answer: Thank you for your comment. This reference was added.

Round 2

Reviewer 1 Report

Thank you for improving the manuscript 

Author Response

Thanks for your comment. Thanks to you and the other reviewers, the paper could be improved.